# Semantic Editing Increment Benefits Zero-Shot Composed Image Retrieval

## ABSTRACT

Zero-Shot Composed Image Retrieval (ZS-CIR) has attracted more attention in recent years, focusing on retrieving a specific image based on a query composed of a reference image and a relative text without training samples. Specifically, the relative text describes the differences between the two images. Prevailing ZS-CIR methods employ image-to-text (I2T) models to convert the query image into a single caption, which is further merged with the relative text by text-fusion approaches to form a composed text for retrieval. However, these methods neglect the fact that ZS-CIR entails considering not only the final similarity between the composed text and retrieved images but also the semantic increment during the compositional editing process. To address this limitation, this paper proposes a training-free method called Semantic Editing Increment for ZS-CIR (SEIZE) to retrieve the target image based on the query image and text without training. Firstly, we employ a pre-trained captioning model to generate diverse captions for the reference image and prompt Large Language Models (LLMs) to perform breadth compositional reasoning based on these captions and relative text, thereby covering the potential semantics of the target image. Then, we design a semantic editing search to incorporate the semantic editing increment contributed by the relative text into the retrieval process. Concretely, we comprehensively consider relative semantic increment and absolute similarity as the final retrieval score, which is subsequently utilized to retrieve the target image in the CLIP feature space. Extensive experiments on three public datasets demonstrate that our proposed SEIZE achieves the new state-of-the-art performance. The code is publicly available at https://anonymous.4open.science/r/SEIZE-11BC.

## CCS CONCEPTS

• **Information systems → Image search**.

## KEYWORDS

Composed image retrieval, zero-shot learning, multi-modal retrieval

**ACM Reference Format:**
Anonymous Author(s). 2022. Semantic Editing Increment Benefits Zero-Shot Composed Image Retrieval. In *Proceedings of (ACM MM)Proceedings of the 32nd ACM International Conference on Multimedia (MM'24), October 28-November 1, 2024, Melbourne, Australia.* ACM, New York, NY, USA, 10 pages. https://doi.org/10.1145/1122445.1122456

**Unpublished working draft. Not for distribution.**

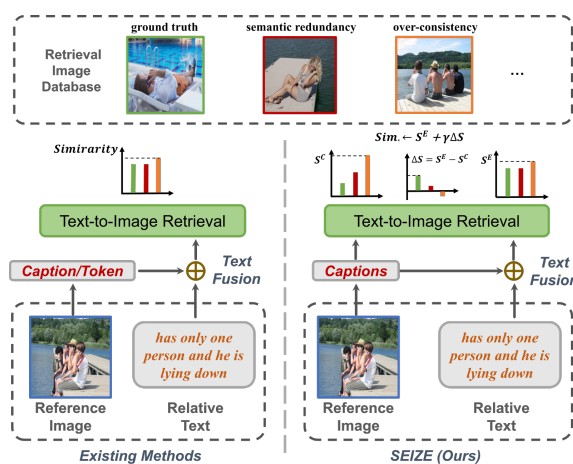

**Figure 1: Comparison of SEIZE with existing methods on ZS-CIR.** SEIZE emphasizes that: (1) The target image is not necessarily the most similar to the composed text in CLIP space and (2) Both semantic editing increment and final similarity matter equally.

## 1 INTRODUCTION

Composed Image Retrieval (CIR) [6, 49, 51] expands upon traditional image retrieval tasks by incorporating natural language descriptions into the image retrieval process. It enables users to search for images based on visual attributes and specify particular alterations to the query image using textual descriptions. The task involves a complex blend of visual content and text modifications to retrieve a novel image that accurately reflects the query's specifics. Previous studies necessitate carefully curated triplets of a reference image, relative text, and a target image, used as training data for a specialized CIR model. However, annotating these triplets is both labor-intensive and challenging. To address this, the recent Zero-Shot Composed Image Retrieval (ZS-CIR) task [4, 41, 43] emphasizes the generalization of CIR models, without requiring annotated triplets, while still focusing on retrieval accuracy.

To carry out ZS-CIR, most works leverage the cross-modal alignment capabilities of large-scale pre-trained Vision-Language Models (VLMs) (e.g., CLIP [40]) to extract aligned features from images and text for matching. Several textual inversion methods [4, 15, 20, 41] have focused on mapping images into pseudo text tokens using image-caption pairs. Then, a static template merges the tokens and the relative text to form target captions. Nonetheless, the reliance of these methods on training images could limit their real-world adaptability and performance. Moreover, the pseudo tokens are often laden with an abundance of visual features, leading to an excessively detailed caption, filled with content, texture, and style specific to the reference image. As illustrated in Figure 1, the target image does not necessarily align strictly with the reference image, and this overemphasis on visual consistency negatively affects retrieval performance. Unlike textual inversion techniques, recent

research has ventured into using Large Language Models (LLMs) for combination. Karthik et al. [27] introduce a simple method involving using a pre-trained image captioning model to caption the reference image. This caption is then recomposed based on the relative text by an LLM to enhance the subsequent retrieval process. In conclusion, the existing methods can be divided into three steps: First, I2T models convert the reference image into a single content text. Second, text-fusion approaches combine the content text with the relative text. Third, the composed caption is projected into the CLIP feature space for target image retrieval.

Previous methods presume that the target image is the one most similar to the composed text in CLIP space, relying heavily on the robustness of I2T models and text-fusion approaches. However, this assumption might be flawed in practical applications due to the inherent limitations of these techniques. Given that: (1) The information provided by the reference image often contains noise, which could misguide the retrieval process by introducing irrelevant details, and (2) The relative text carries high-quality, dense information, all of which is relevant and beneficial to the retrieval process. Prior methods often neglect a critical aspect: Despite the target image not being highly similar to the reference image or its text conversion, compositional editing of the reference image and the relative text significantly improves their similarity or ranking. As exemplified in Figure 1, the ground truth in green does not achieve the highest similarity when compared with the combined text of the reference image and the relative text. This is due to the absence of the concept of 'dock' within the ground truth. However, the similarity increment before and after the combination with the relative text is the highest. Inspired by the above observation, we aim to address **Challenge 1**: How to represent and incorporate the semantic editing increment contributed by the relative text into the retrieval process?

Furthermore, most existing methods overlook that ZS-CIR is essentially a *fuzzy matching* [9, 12, 39] task, where the semantics of the target image are not strictly determined by the reference image and relative text. This ambiguity stems from the diverse nature of the multi-modal query in ZS-CIR. The relative text aims to modify the semantics of the reference image, but the query does not explicitly specify which visual objects/attributes to modify, retain, or omit. Therefore, as shown in Figure 1, this semantic ambiguity can result in a wide range of possible semantics of the retrieval target. For example, the relative text does not clarify whether the person in the target image should be on the dock or whether he should be talking on the phone. However, LLM may describe a person on the dock who is not talking on the phone in the target image, which introduces semantic bias. Hence, generating just one edited caption may fail to adequately capture the diversity of potential composed results, leading to suboptimal retrieval performance for ZS-CIR. From the preceding discussion, we have to tackle **Challenge 2**: How to effectively generate a variety of edited captions that encompass diverse potential semantics of the composed results?

To deal with the above challenges, we propose a novel **S**emantic **E**diting **I**ncrement for **ZE**ro-shot composed image retrieval (**SEIZE**) method, retrieving a composed image based on a reference image and a relative text without training. For **Challenge 1**, we propose a semantic editing search to comprehensively consider both the increment of semantic editing and the final similarity. Our approach draws inspiration from *formative assessment* [7, 8, 14] extensively studied in education, which refers to an assessment process, emphasizing not only the end result but also the improvement facilitated by the teaching process. Therefore, we separately calculate the cosine similarity between the composed text and the content text with the retrieval images. We compute the semantic editing increment as the difference in similarity or ranking, add it to the absolute similarity for the final adjusted score, and then retrieve the composed image based on this score in the CLIP feature space. For **Challenge 2**, we propose a breadth compositional reasoning to generate diverse edited captions, covering possible semantics of the composed results for fuzzy retrieval. To conduct it, we generate diverse captions for the reference image focusing on different semantic perspectives. Then, we prompt an LLM to infer breadth edited captions based on the relative text, describing possible composed images of diverse semantics. Extensive experiments on three benchmark datasets for ZS-CIR indicate significant performance improvements in our proposed SEIZE compared with the state-of-the-art methods.

In summary, our contributions can be summarized as follows:

- We propose a novel **S**emantic **E**diting **I**ncrement for **ZE**ro-shot composed image retrieval (**SEIZE**) method, which can utilize off-the-shelf tools to accurately retrieve a specific image based on a reference image and a relative text without training.
- We propose a plug-and-play semantic editing search to incorporate the semantic editing increment contributed by the relative text into the retrieval process, which is simple yet effective, seamlessly enhancing various methods for ZS-CIR.
- We propose an LLM-based breadth compositional reasoning to generate diverse edited captions from different semantic perspectives of the reference image. The diverse captions edited by LLMs can effectively cover the possible semantics of the composed results, overcoming the fuzzy nature of ZS-CIR.
- Extensive experiments conducted on three benchmark datasets demonstrate significant performance improvements of the proposed SEIZE compared with both the state-of-the-art training-dependent and training-free methods for ZS-CIR.

## 2 RELATED WORK

### 2.1 Composed Image Retrieval

Composed Image Retrieval (CIR) task [11, 18, 29, 47], focusing on retrieving a target image based on a query of a reference image and a relative text, has gained significant attention in recent years. CIR combines compositional learning [25, 28] with image retrieval, forming a distinctive and demanding task, which has been widely applied in conditional search [6] and fashion styling [50]. For instance, Text-Image Residual Gating [47], leverages ResNet [22] for image feature extraction and LSTM [23] for text feature extraction. Then, a residual module combines these multi-modal query features to produce the composed feature. An extension [11] introduces an attention module to blend hierarchical reference features with relative text features. However, those CIR methods rely on annotated data, which is complicated and requires extensive labor.

Therefore, Zero-Shot CIR (ZS-CIR) [4, 41, 43] has recently garnered significant interest, aiming to design generalized CIR models without annotated data. Existing ZS-CIR methods commonly transform the image modality into a text modality using methods like

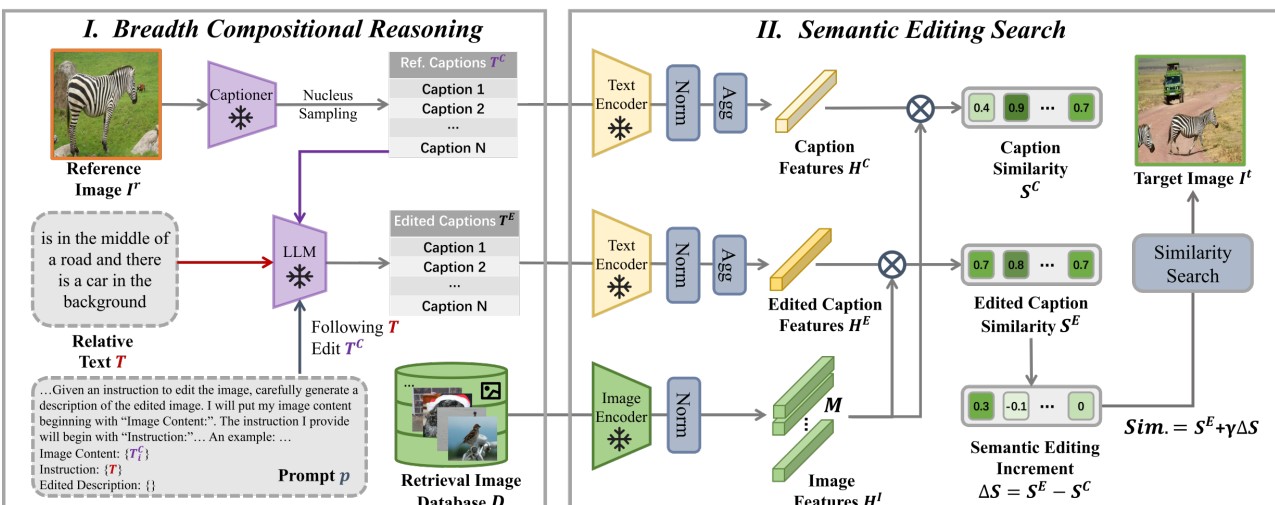

**Figure 2: Architecture of the proposed SEIZE method:** (1) Breadth compositional reasoning is designed to generate diverse edited captions from different semantic perspectives of the reference image in response to the fuzzy nature of ZS-CIR; (2) Semantic editing search is designed to incorporate the semantic editing increment contributed by the relative text into the retrieval process for the final retrieval.

a captioning model or textual inversion. Some existing methods [4, 41] use image-caption pairs to train textual inversions mapping images to text tokens. A static template merges tokens and relative text to obtain target captions, performing CIR without explicit supervision based on CLIP [40]. Recently, Karthik et al. [27] propose a training-free method that captions the reference image using a pre-trained VLM and employs an LLM to recompose the caption based on the relative text for retrieval. However, they solely consider the final similarity between the composed text and retrieved images for matching, overlooking the fact that the target image is not necessarily the most similar to the composed text in CLIP space. To overcome this problem, we propose a plug-and-play semantic editing search to incorporate the semantic editing increment contributed by the relative text into the retrieval process.

### 2.2 Vision-Language Models for CIR

The popularity of the pre-trained BERT [17] model has sparked interest in developing pre-trained Vision-Language Models (VLMs), including [13, 32, 33, 37, 42], aiming to create Transformer-based [45] models trained on large-scale image-text triplets to produce vision-and-language representations. For CIR, to map images and text into a shared embedding space, many methods harness large pre-trained multi-modal models, such as CLIP [40], as the backbone for feature extraction. These models [5, 21, 27] have recently gained popularity due to their exceptional ability to handle multi-modal data. For example, Baldrati et al. [5] utilize CLIP to extract both image and text features and then employ a combiner module to merge the multi-modal query features, achieving excellent retrieval performance. Recently, Han et al. [21] further advance this approach by designing a unified visual-language model capable of managing multiple multi-modal learning tasks, including CIR. They use various cross-attention adaptors and achieve leading performance in CIR by leveraging the large model and multi-task learning. Recent progress has been made with models such as BLIP [31] and OFA [48], which move beyond shared space projection to address

various vision-language tasks, such as captioning [46] and visual question answering [3]. While these models have relied on these models indirectly for CIR via specialized modules [6, 16, 47] and fine-tuning [19], our research shows that when vision-language models are combined with an LLM, they can effectively perform CIR without requiring additional training.

## 3 PROBLEM STATEMENT

**Zero-Shot Composed Image Retrieval (ZS-CIR).** Composed Image Retrieval (CIR) can be defined as a multi-modal retrieval problem. Given a reference image $I^r$ and relative text $T$, the objective is to retrieve the target image $I^t$ from an image database $\mathcal{B}$ that aligns with the relative text while preserving the underlying semantic content of the image that has not been explicitly mentioned. Zero-shot CIR further entails the absence of training samples, which should be conducted with off-the-shelf tools.

## 4 METHOD

As stated in the problem statement, the input of CIR is a multi-modal query $q = \{I^r, T\}$, where $I^r$ and $T$ denote the reference image and the relative text, respectively. We leverage a combination of visual and textual information for retrieval. The reference image serves as the visual feature, while the relative text provides additional context or constraints for the retrieval process. Our objective is to find the target image that not only matches the relative text but also captures the essence of the reference image. To this end, we propose a method called Semantic Editing Increment for ZS-CIR (**SEIZE**), which consists of breadth compositional reasoning and semantic editing search, as depicted in Figure 2.

### 4.1 Breadth Compositional Reasoning

To capture comprehensive coverage of potential semantics in the composed results, we introduce a method based on LLMs, which facilitates broad-scope compositional reasoning to generate diverse edited captions as shown in Figure 3. The method comprises a

multi-caption generator for crafting multiple reference captions and a multi-prompt editing reasoner for reasoning and editing.

*4.1.1 Multi-caption Generator.* Unlike earlier methods [4, 20, 41] that directly extract visual features from reference images, our approach for CIR uses a language-for-vision strategy. Rather than relying on visual features, we harness a pre-trained captioning model $\phi(\cdot)$, such as OFA [48], Flamingo [2] or BLIP-2 [30], to generate natural language descriptions of the reference image. The method for generating multiple captions can be described as follows:

$$\mathcal{T}^C = \{T_i^C \mid 0 \le i < N\} = \phi(I^r), \tag{1}$$

where $T_i^C$ represents the $i$-th caption generated for the reference image, and $N$ is the total count of captions produced. To ensure the generation of diverse captions and prevent repetition, we employ nucleus sampling [24] during the captioning process, improving the diversity of the generated captions.

*4.1.2 LLM-based Editing Reasoner.* Relying solely on either relative text or reference captions alone fails to provide complete context and modification details. While predefined templates like "a photo of {REF} that {REL}", can fixedly combine them, they lack the adaptability to accommodate diverse forms of relative text and determine the most suitable caption format for a specific query. To tackle this, we leverage the reasoning capabilities of existing LLMs. Instead of merging the reference caption and relative text by a static template, we derive cohesive, unified, and breadth captions edited by LLMs.

Formally, given the diverse captions $\mathcal{T}^C$ of the reference image and relative text $T$, we design a simple prompt template $f(\cdot, \cdot)$ inspired by [35], combining the reference captions and relative text to create the full prompts $\mathcal{P}$ for LLM:

$$\mathcal{P} = \{p_i = f(T_i^C, T) \mid 0 \le i < N\}, \tag{2}$$

where $p_i$ represents the prompt that combines the reference image caption and relative text. The image caption serves as a content prompt prepended with "Image Content:", and the relative text serves as a modification instruction prepended with "Instruction:", as shown in Figure 2. We fill these two parts into the template to get the full prompt. Then, we input the prompts into the LLM for reasoning and obtain the generated edited captions $\mathcal{T}^E$:

$$\mathcal{T}^E = \{T_i^E = \text{LLM}(p_i) \mid 0 \le i < N\}. \tag{3}$$

We include only one example in each LLM query for compositional reasoning to generate a variety of breadth-edited captions efficiently and cost-effectively. Refer to the **Supplementary** for a detailed description of prompt template $f(\cdot, \cdot)$.

## 4.2 Semantic Editing Search

To effectively incorporate the semantic increment contributed by the relative text into the retrieval process, we propose a semantic editing search to comprehensively consider relative semantic increment and absolute similarity as the final retrieval score.

*4.2.1 Feature Extractor.* To retrieve the target image from the image database using edited captions, we need to extract features from both modalities and align them in a common feature space. For this purpose, we use the image and text encoders from large-scale

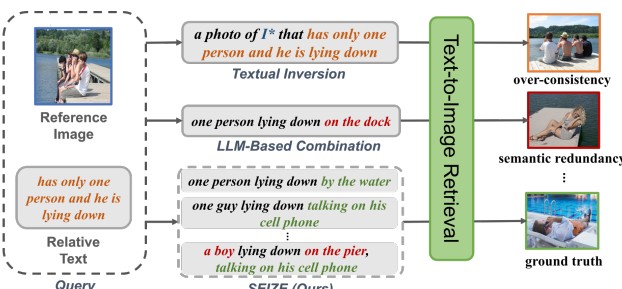

**Figure 3: Illustration of the breadth compositional reasoning.**

VLMs (e.g. CLIP) to process the image database and edited captions, respectively.

**Visual Feature Extractor:** Given the image-search database $\mathcal{B}$, for each image $I_i \in \mathbb{R}^{H \times W \times 3}$ in the database, we use the CLIP image encoder $\psi_I(\cdot)$ to extract the image feature and normalize it to obtain $f_i^I \in \mathbb{R}^d$, where $d$ is the dimension of the joint feature space. Formally, we extract the visual features $\mathcal{F}^I \in \mathbb{R}^{M \times d}$ of the entire image database as follows:

$$\mathcal{F}^I = \{f_i^I = \frac{\psi_I(I_i)}{\|\psi_I(I_i)\|_2} \mid 0 \le i < M\}, \tag{4}$$

where $M = |\mathcal{B}|$ is the number of candidate images in the database.

**Textual Feature Extractor:** Given the edited captions $\mathcal{T}^E$ and CLIP text encoder $\psi_T(\cdot)$, we extract the textual features $\mathcal{F}^E \in \mathbb{R}^{N \times d}$ of edited captions as follows:

$$\mathcal{F}^E = \{f_i^E = \frac{\psi_T(T_i^E)}{\|\psi_T(T_i^E)\|_2} \mid 0 \le i < N\}. \tag{5}$$

Then, we aggregate the textual features of edited captions into a unified feature as $f^E = \text{Agg}(\mathcal{F}^E)$. Similarly, we extract the textual features $\mathcal{F}^C$ of diverse captions $\mathcal{T}^C$, and the corresponding unified feature $f^C$ for subsequent calculations.

*4.2.2 Semantic Editing Increment.* To quantify the significance of semantic improvement in the similarity between text and images after the compositional editing of the reference image and the relative text, we employ a measure of semantic increment before and after editing as the scoring metric for images in the database. In Figure 4, the relative text guides the reference image toward the target image to form an edited reference caption. However, the image most similar to the edited caption is not the target image, but the hard negative image. In this case, where traditional retrieval methods fail, the semantic editing increment is denoted by a purple bidirectional arrow. Leveraging the principle that in a triangle, the difference between any two sides is less than the length of the third, it becomes evident that the reference image yields the maximum semantic editing increment to the target image among all retrieval images. Note that for the sake of a more intuitive example, we employ distance metrics. However, the same concept can be seamlessly applied to similarity metrics as well.

Formally, given the aggregated feature $f^C$ of reference captions and the feature $f^E$ of their corresponding edited captions, we independently calculate their similarity to the visual features $\mathcal{F}^I$ as

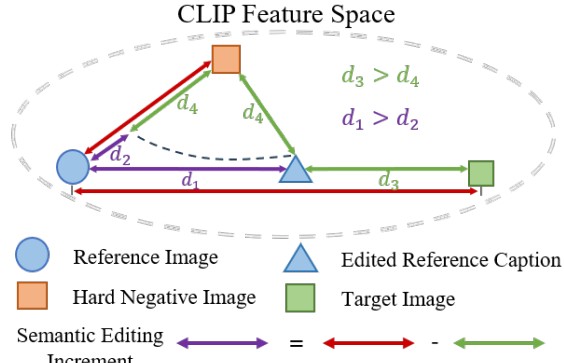

**Figure 4: Illustration of semantic editing increment.** Relative text guides the reference image toward the target image to form an edited reference caption. Hard negative image where $d_3 > d_4$ harms retrieval, while semantic editing increment improves it as $d_1 > d_2$.

follows:

$$S^C = f^C(\mathcal{F}^I)^T,$$
$$S^E = f^E(\mathcal{F}^I)^T, \tag{6}$$

where $S^C, S^E \in \mathbb{R}^M$ represent the cosine similarities between the reference and edited captions, respectively, and all $M$ retrieval images. Here, $S_i \in [-1, 1]$ since all visual and text features are preconditioned by $L_2$ normalization in the CLIP space. Then, we calculate their difference $\Delta S = S^E - S^C$ as the semantic editing increment. Finally, we calculate the adjusted similarity $S^A$ as follows:

$$S^A = S^E + \alpha[\Delta S]_{>0} + \beta[\Delta S]_{<0}, \tag{7}$$

where $\alpha$ represents the weight factor for the positive semantic editing increment, and $\beta$ represents the weight factor for the negative semantic editing increment. Indeed, we can also use ranking changes as the semantic editing increment to calculate the adjusted similarity, with comparison results as shown in Section 5.4.

*4.2.3 Similarity Search.* Based on the adjusted similarity between the image features in the database and the feature of edited captions, we make the following calculation and search:

$$k = \arg\max_i S_i^E, \tag{8}$$

where $k$ is the index of the target image in the database and we get the final target retrieved image $I^t = \mathcal{B}(k)$.

## 5 EXPERIMENTS

We conduct extensive experiments to investigate the effectiveness of our proposed method for composed image retrieval.

### 5.1 Experimental Setup

*5.1.1 Datasets.* We compare SEIZE with state-of-the-art baselines on three public datasets: CIRCO [4], CIRR [36], and FashionIQ [50], which have been widely used for CIR. CIRR is the first natural image dataset designed specifically for CIR. There are 36,554 queries, each containing a single target image. CIRCO is based on real-world images from the COCO 2017 unlabeled set [34] and provides multiple ground truths. It provides 123,403 images for retrieval

and 4.53 ground truths per query on average, which offers a more robust evaluation for CIR models. FashionIQ focuses on the fashion domain and is divided into three subcategories: Dress, Shirt, and Toptee. It comprises 30,135 triplets for query and 77,683 images for retrieval. In particular, we employ the test sets of CIRR and CIRCO and three categories of FashionIQ validation split for ZS-CIR.

*5.1.2 Evaluation Metrics.* On CIRCO, given that each query has multiple target images, we utilize mean Average Precision (mAP), a more fine-grained metric to consider the rank of retrieval results. Specifically, we apply mAP@k for our evaluation, where $k \in \{5, 10, 25, 50\}$ represents the number of top-ranked retrieval results under consideration. On CIRR, we use Recall@k ($k \in \{1, 5, 10\}$) as the main metric, which denotes the percentage of target images included in the top-k list. Besides, we also evaluate with the subset setting, where there are only 6 most similar images (one ground truth) within the subset of the query in the database, denoted as Recall$_{Subset}$k ($k \in \{1, 2, 3\}$). On FashionIQ, we also employ Recall@k ($k \in \{10, 50\}$) as the main metric. We separately compute the Recalls for three categories (Dress, Shirt, and Toptee) from the FashionIQ validation split, and then calculate the average of them.

*5.1.3 Implementation Details.* For the captioning model, we use the pre-trained BLIP-2 [30] with a large language model OPT-6.7b. We employ nucleus sampling [24] during the caption generation process to generate diverse captions, and we set the number $N$ of captions to 15. For VLM, we use the ViT-B/32 and ViT-L/14 CLIP [40] from OpenAI, as well as the ViT-H/14 and ViT-G/14 CLIP from OpenCLIP[26]. For LLM, we use gpt-3.5-turbo [10] by default, but we also conduct extensive experiments with Llama2-70B [44] and GPT-4 [1]. The weight factors $\alpha$ and $\beta$ in the SES module are set to 0.13 and 2.1. The feature space dimensions are as follows: 512 for B/32, 768 for L/14, and 1024 for H/14 and G/14. The whole model is implemented by Pytorch [38] with one NVIDIA A100 GPU.

### 5.2 Baselines

We select the following state-of-the-art baselines for ZS-CIR to conduct a comprehensive comparison. *Image-only*: Only the features of reference images, which are extracted by the CLIP image encoder, are used to compute similarity for retrieval; *Text-only*: Only the relative text features, extracted by the CLIP text encoder, are utilized as retrieval features to calculate similarity; *Captioning*: We employ BLIP-2 to generate captions of reference images for retrieval; *PALAVRA* [15]: A two-stage approach based textual inversion with a mapping function and a subsequent optimization of the pseudo-word token; *Pic2Word* [41]: A training-dependent method employs a textual inversion network optimized by contrastive loss to capture the pseudo-word token for retrieval; *SEARLE* [4]: A training-dependent method where pseudo-word tokens of images are generated with a textual inversion and then distill their knowledge to a textual inversion network; *SEARLE-OTI* [4]: A variant of SEARLE without the distillation network; *LinCIR* [20]: A self-supervision method is used for training, which projects text embeddings into the token space for retrieval; *CIReVL* [27]: A training-free method using a generative VLM and asking an LLM to recompose the caption based on the textual modification for retrieval.

**Table 1: Results of comparison among different models on CIRCO and CIRR test sets.** Best and second-best scores are highlighted in bold and underlined, respectively.

| Backbone | Method | Training-free | CIRCO | | | | CIRR | | | | | |
|---|---|---|---|---|---|---|---|---|---|---|---|---|
| | | | mAP@k | | | | Recall@k | | | Rs@k | | |
| | | | k=5 | k=10 | k=25 | k=50 | k=1 | k=5 | k=10 | k=1 | k=2 | k=3 |
| ViT-B/32 | PALAVRA(ECCV'22) | ✗ | 4.61 | 5.32 | 6.33 | 6.80 | 16.62 | 43.49 | 58.51 | 41.61 | 65.30 | 80.94 |
| | SEARLE(ICCV'23) | ✗ | 9.35 | 9.94 | 11.13 | 11.84 | 24.00 | 53.42 | 66.82 | 54.89 | 76.60 | 88.19 |
| | SEARLE-OTI(ICCV'23) | ✗ | 7.14 | 7.83 | 8.99 | 9.60 | 24.27 | 53.25 | 66.10 | 54.10 | 75.81 | 87.33 |
| | CIReVL(ICLR'24) | ✔ | 14.94 | 15.42 | 17.00 | 17.82 | 23.94 | 52.51 | 66.00 | 60.17 | 80.05 | 90.19 |
| | SEIZE(Ours) | ✔ | **19.04** | **19.64** | **21.55** | **22.49** | **27.47** | **57.42** | **70.17** | **65.59** | **84.48** | **92.77** |
| ViT-L/14 | Image-only | ✔ | 1.28 | 1.70 | 2.35 | 2.69 | 3.64 | 12.75 | 23.32 | 11.58 | 31.41 | 45.26 |
| | Text-only | ✔ | 2.63 | 2.85 | 3.30 | 3.58 | 20.51 | 43.21 | 55.08 | 60.39 | 80.02 | 90.05 |
| | Captioning | ✔ | 1.65 | 1.96 | 2.42 | 2.71 | 4.05 | 15.88 | 25.69 | 20.87 | 40.60 | 60.89 |
| | Pic2Word(CVPR'23) | ✗ | 8.72 | 9.51 | 10.64 | 11.29 | 23.90 | 51.70 | 65.30 | 53.76 | 74.46 | 87.08 |
| | SEARLE(ICCV'23) | ✗ | 11.68 | 12.73 | 14.33 | 15.12 | 24.24 | 52.48 | 66.29 | 53.76 | 75.01 | 88.19 |
| | SEARLE-OTI(ICCV'23) | ✗ | 10.18 | 11.03 | 12.72 | 13.67 | 24.87 | 52.31 | 66.29 | 53.80 | 74.31 | 86.94 |
| | LinCIR(CVPR'24) | ✗ | 12.59 | 13.58 | 15.00 | 15.85 | 25.04 | 53.25 | 66.68 | 57.11 | 77.37 | 88.89 |
| | CIReVL(ICLR'24) | ✔ | 18.57 | 19.01 | 20.89 | 21.80 | 24.55 | 52.31 | 64.92 | 59.54 | 79.88 | 89.69 |
| | SEIZE(Ours) | ✔ | **24.98** | **25.82** | **28.24** | **29.35** | **28.65** | **57.16** | **69.23** | **66.22** | **84.05** | **92.34** |
| ViT-G/14 | Pic2Word(CVPR'23) | ✗ | 5.54 | 5.59 | 6.68 | 7.12 | 30.41 | 58.12 | 69.23 | 68.92 | 85.45 | 93.04 |
| | SEARLE(ICCV'23) | ✗ | 13.20 | 13.85 | 15.32 | 16.04 | 34.80 | 64.07 | 75.11 | 68.72 | 84.70 | 93.23 |
| | LinCIR(CVPR'24) | ✗ | 19.71 | 21.01 | 23.13 | 24.18 | 35.25 | 64.72 | 76.05 | 63.35 | 82.22 | 91.98 |
| | CIReVL(ICLR'24) | ✔ | 26.77 | 27.59 | 29.96 | 31.03 | 34.65 | 64.29 | 75.06 | 67.95 | 84.87 | 93.21 |
| | SEIZE(Ours) | ✔ | **32.46** | **33.77** | **36.46** | **37.55** | **38.87** | **69.42** | **79.42** | **74.15** | **89.23** | **95.71** |

Among them, *Image-only*, *Text-only*, and *Captioning* are simple technologies directly applied to one modality; *PALAVRA*, *Pic2Word*, *SEARLE*, *SEARLE-OTI*, and *LinCIR* are training-dependent methods based on textual inversion; *CIReVL* is the state-of-the-art training-free method. For a fair comparison, the results of the published baselines are derived from their original papers.

## 5.3 Results and Analysis

**CIRCO:** The left section of Table 1 displays CIRCO test results. Based on them, we have the following observations: (1) Among the simpler baselines, *Image-only* and *Captioning* perform worse than *Text-only*, indicating the importance of relative text for CIR. Besides, *Captioning* outperforms *Image-only*, suggesting that textual features from image captions are more suitable for CIR than direct visual features. (2) Among I2T-based methods, the methods based on pre-trained captioning models, *CIReVL* and *SEIZE*, perform better than the methods based on pseudo-word, *PALAVRA*, *Pic2Word*, *SEARLE*, and *LinCIR*. This demonstrates that the captions generated by the captioning model possess semantics that are more suitable for CLIP text encoding compared to textual inversion. (3) *SEIZE* consistently outperforms all baselines across all metrics and CLIP backbones. For instance, using ViT-L/14, SEIZE outperforms the second-best *CIReVL* by 34.52% in mAP@5, 35.82% in mAP@10, 35.18% in mAP@25, and 34.63% in mAP@50, proving the effectiveness of SEIZE for CIR.

**CIRR:** The right section of Table 1 presents CIRR test results. From these results, key observations include: (1) *Text-only* performs significantly better than *Image-only* and *Captioning*, indicating a minimal correlation between reference and target images due to the noisy dataset and the lesser information provided by reference

images. (2) Despite the noise, *SEIZE* consistently outperforms all baselines across all CLIP backbones. This highlights the robustness and adaptability of *SEIZE* even in noisy data and diverse scenarios. (3) Using ViT-L/14 CLIP, *SEIZE* surpasses the second-best method by 14.42% in Recall@1, 7.34% in Recall@5, and 3.82% in Recall@10, emphasizing the effectiveness of *SEIZE*.

**FashionIQ:** Table 2 shows the results from the FashionIQ validation set. Key observations include: (1) *Text-only* methods perform better than *Image-only* methods, but not significantly better than the *Captioning* method. This suggests that while reference images lack key information, converting them into captions is empirically a better strategy, possibly due to the subtle differences in style, color, or pattern in fashion images that are hard to capture through visual features directly. (2) *SEIZE* outperforms all baselines in all metrics with ViT-B/32 and L/14 CLIP. When using L/14, *SEIZE* relatively outperforms the second-best *CIReVL* by 16.22% in average Recall@10 and 11.61% in average Recall@50. However, *LinCIR* achieves the best average R@10 on ViT-G/14, likely due to superior supervised optimization of training-based methods on larger VLMs. Despite this, *SEIZE* remains the top-performing training-free method.

## 5.4 Ablation Study

In this section, we conduct ablation studies to evaluate the individual contributions of components in our method. To avoid confusion, we assess the three main components separately by comparing SEIZE variants. The following three categories of SEIZE variants are designed for comparative analysis:

- **Breadth Compositional Reasoning (BCR) module:** (1) A variant of SEIZE without LLMs and only using a simple template "a photo of {caption} that {relative text}", (2) a variant without

**Table 2: Results of comparison among different models on FashionIQ validation set.** Best and second-best scores are highlighted in bold and underlined, respectively.

| Backbone | Method | Training-free | Shirt | | Dress | | Toptee | | Average | |
|---|---|---|---|---|---|---|---|---|---|---|
| | | | R@10 | R@50 | R@10 | R@50 | R@10 | R@50 | R@10 | R@50 |
| ViT-B/32 | Image-only | ✔ | 6.92 | 14.23 | 4.46 | 12.19 | 6.32 | 13.77 | 5.90 | 13.37 |
| | Text-only | ✔ | 19.87 | 34.99 | 15.42 | 35.05 | 20.81 | 40.49 | 18.70 | 36.84 |
| | Captioning | ✔ | 17.47 | 30.96 | 9.02 | 23.65 | 15.45 | 31.26 | 13.98 | 28.62 |
| | PALAVRA(ECCV'22) | ✘ | 21.49 | 37.05 | 17.25 | 35.94 | 20.55 | 38.76 | 19.76 | 37.25 |
| | SEARLE(ICCV'23) | ✘ | 24.44 | 41.61 | 18.54 | 39.51 | 25.70 | 46.46 | 22.89 | 42.53 |
| | SEARLE-OTI(ICCV'23) | ✘ | 25.37 | 41.32 | 17.85 | 39.91 | 24.12 | 45.79 | 22.44 | 42.34 |
| | CIReVL(ICLR'24) | ✔ | 28.36 | 47.84 | 25.29 | 46.36 | 31.21 | 53.85 | 28.29 | 49.35 |
| | SEIZE(Ours) | ✔ | **29.38** | **47.97** | **25.37** | **46.84** | **32.07** | **54.78** | **28.94** | **49.86** |
| ViT-L/14 | Pic2Word(CVPR'23) | ✘ | 26.20 | 43.60 | 20.00 | 40.20 | 27.90 | 47.40 | 24.70 | 43.70 |
| | SEARLE(ICCV'23) | ✘ | 26.89 | 45.58 | 20.48 | 43.13 | 29.32 | 49.97 | 25.56 | 46.23 |
| | SEARLE-OTI(ICCV'23) | ✘ | 30.37 | 47.49 | 21.57 | 44.47 | 30.90 | 51.76 | 27.61 | 47.90 |
| | LinCIR(CVPR'24) | ✘ | 29.10 | 46.81 | 20.92 | 42.44 | 28.81 | 50.18 | 26.28 | 46.49 |
| | CIReVL(ICLR'24) | ✔ | 29.49 | 47.40 | 24.79 | 44.76 | 31.36 | 53.65 | 28.55 | 48.57 |
| | SEIZE(Ours) | ✔ | **33.04** | **53.22** | **30.93** | **50.76** | **35.57** | **58.64** | **33.18** | **54.21** |
| ViT-G/14 | Pic2Word(CVPR'23) | ✘ | 33.17 | 50.39 | 25.43 | 47.65 | 35.24 | 57.62 | 31.28 | 51.89 |
| | SEARLE(ICCV'23) | ✘ | 36.46 | 55.35 | 28.16 | 50.32 | 39.83 | 61.45 | 34.81 | 55.71 |
| | LinCIR(CVPR'24) | ✘ | **46.76** | 65.11 | 38.08 | 60.88 | **50.48** | 71.09 | **45.11** | 65.69 |
| | CIReVL(ICLR'24) | ✔ | 33.71 | 51.42 | 27.07 | 49.53 | 35.80 | 56.14 | 32.19 | 52.36 |
| | SEIZE(Ours) | ✔ | 43.60 | **65.42** | **39.61** | **61.02** | 45.94 | **71.12** | 43.05 | **65.85** |

**Table 3: Ablation study on CIRCO test set with ViT-L/14 CLIP.** - denotes the default setting of SEIZE.

| Abl. | | | CIRCO (mAP@k) | | | |
|---|---|---|---|---|---|---|
| BCR | Agg. | SES | k=5 | k=10 | k=25 | k=50 |
| w/o LLM | - | - | 14.37 | 15.48 | 17.21 | 18.19 |
| w/o Cap.&LLM | w/o | w/o | 2.63 | 2.85 | 3.30 | 3.58 |
| w/o Mul. | w/o | - | 19.67 | 20.56 | 22.43 | 23.44 |
| - | Max. | - | 19.21 | 19.73 | 21.71 | 22.69 |
| | Opt. | | 20.46 | 21.38 | 23.07 | 24.19 |
| - | - | w/o | 21.93 | 22.42 | 24.86 | 25.88 |
| | | Rank | 23.49 | 24.13 | 26.52 | 27.59 |
| **SEIZE** | | | **24.98** | **25.82** | **28.24** | **29.35** |

**Table 4: Impact of SES module on various ZS-CIR methods.**

| Method | SES | CIRCO (mAP@k) | | | | |
|---|---|---|---|---|---|---|
| | | k=5 | k=10 | k=25 | Avg. | ↑ |
| Pic2Word | ✘ | 8.72 | 9.51 | 10.64 | 9.62 | - |
| | ✔ | 10.55 | 11.38 | 12.61 | 11.51 | 19.64% |
| SEARLE | ✘ | 11.63 | 12.70 | 14.30 | 12.88 | - |
| | ✔ | 14.99 | 15.67 | 17.22 | 15.96 | 23.91% |
| LinCIR | ✘ | 12.71 | 13.61 | 14.98 | 13.77 | - |
| | ✔ | 15.53 | 16.59 | 18.05 | 16.72 | 21.42% |
| CIReVL | ✘ | 18.02 | 18.81 | 20.79 | 19.21 | - |
| | ✔ | 19.86 | 20.93 | 22.92 | 21.24 | 10.57% |
| SEIZE(Ours) | ✘ | 21.93 | 22.42 | 24.86 | 23.07 | - |
| | ✔ | 24.98 | 25.82 | 28.24 | 26.35 | 15.72% |

LLMs or captioning models and only using the relative text for retrieval, and (3) a variant without multiple diverse captions and only using a single caption.

- **Aggregate function:** A variant of SEIZE with the maximum feature (the maximum similarity feature for each image), and a variant with the optimal feature (the feature of a single edited caption with the best retrieval result) instead of the average feature of edited captions.
- **Semantic Editing Search (SES) module:** A variant of SEIZE without the SES module, and a variant with ranking increment (integer, multiplied by a weight factor of 0.3) instead of similarity.

The ablation study results are shown in Table 3.

- **Effects of the BCR module:** The results of the BCR column show that: (1) SEIZE outperforms the variant without LLMs (*w/o LLM*) on the CIRCO test set, highlighting the effectiveness of the LLM-based editing reasoner for combining reasoning and editing captions based on relative text. The ability of LLMs to generate natural language leads to engaging and coherent text. It also eliminates semantic irrelevance within diverse captions and improves retrieval accuracy, greatly benefiting SEIZE. (2) The variant *w/o LLM* performs better than *w/o Cap.&LLM* with the fixed relative text, underlining the importance of the caption generator in providing essential reference image details and context. (3) SEIZE outperforms the variant *w/o Mul.*, highlighting the importance of multiple captions. For more details on how the number of captions affects the results, refer to Section 5.5.
- **Effects of the aggregate function:** The results of the *Agg.* column indicate that: The proposed SEIZE outperforms the *Max.* and *Opt.* variants, underscoring the effectiveness of the average function in providing a more complementary and comprehensive retrieval perspective using diverse semantic information.

**Table 5: Impact of different VLMs and LLMs.**

| VLM | LLM | CIRCO (mAP@k) | | | |
|---|---|---|---|---|---|
| | | k=5 | k=10 | k=25 | k=50 |
| ViT-L/14 | - | 14.35 | 15.64 | 17.24 | 18.03 |
| | LLama2-70B | 22.28 | 23.41 | 25.84 | 26.96 |
| | GPT-3.5-Turbo | 24.98 | 25.82 | 28.24 | 29.35 |
| | GPT-4 | 25.12 | 26.12 | 28.67 | 29.72 |
| ViT-B/32 | GPT-3.5-Turbo | 19.04 | 19.64 | 21.55 | 22.49 |
| ViT-L/14 | | 24.98 | 25.82 | 28.24 | 29.35 |
| ViT-H/14 | | 32.07 | 33.49 | 36.23 | 37.31 |
| ViT-G/14 | | 32.46 | 33.77 | 36.46 | 37.55 |

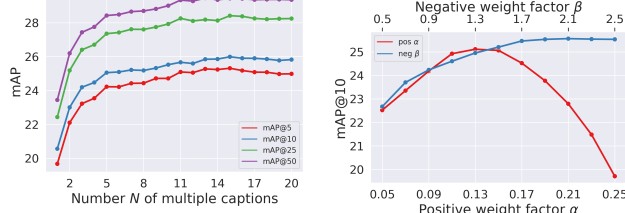

**Figure 5: Sensitivity of different hyperparameters on CIRCO.**

- **Effects of the SES module:** The results of the SES column show that: SEIZE outperforms the variants with ranking scores and without SES, highlighting the effectiveness of the semantic editing increment for additional consideration of improvements by compositional reasoning. For more details on the impact of SES on various ZS-CIR methods, refer to Section 5.6.

## 5.5 Hyperparameter Analysis

To analyze the sensitivity of the hyperparameter in SEIZE, we conduct experiments with controlled variables on the CIRCO test set. We use pre-trained BLIP-2 for captioning, ViT-L/14 CLIP for image and text feature extraction, and gpt-3.5-turbo for reasoning.

**Number $N$ in the Multi-caption Generator:** To examine the impact of the number $N$ of samples for the captioning model, we vary the value of $N$ from 1 to 20 and report the results on the CIRCO test set in Figure 5. The results across the four metrics show a pattern: an initial sharp increase followed by relative stability as $N$ increases. The model achieves a good balance between computational cost and effect when $N$ is between 13 and 16.

**Weight factors $\alpha$ and $\beta$ in SES module:** We vary $\alpha$ from 0.05 to 0.25 and $\beta$ from 0.5 to 2.5 to analyze the influence of weight factors. As $\alpha$ increases, results first rise and then fall, while as $\beta$ increases, results initially rise and then stabilize. From Figure 5, we can observe that the model achieves the best performance with $\alpha = 0.13$ and $\beta = 2.1$. The negative semantic editing increment is more critical due to its larger weight factor, which maintains fine results as it increases. However, SES is sensitive to the positive semantic editing increment, and increasing its weight leads to poor performance. This aligns with intuition: images matching the semantics of relative text may not necessarily be the target image, but those that deviate from it are, with high probability, not the target image.

## 5.6 Impact of SES module on ZS-CIR Methods

Considering that current ZS-CIR methods utilize I2T models to transform the query image into a caption, which is then combined with the relative text via text-fusion approaches to generate a final text for retrieval, it is plausible to integrate the Semantic Editing Search (SES) module into various ZS-CIR methods that rely on the language-for-vision strategy. The experimental results and improvements before and after using the SES module on ZS-CIR methods with ViT-L/14 CLIP are shown in Table 4. It can be observed that the SES module has improved all methods, especially the textual inversion methods, with an average improvement of 21.66%. This proves that our SES module is plug-and-play, simple yet effective, allowing for seamless integration with various methods for ZS-CIR.

## 5.7 Impact of VLMs and LLMs

Since our proposed SEIZE relies heavily on VLMs for retrieval and LLMs for reasoning, we study several VLMs and LLMs to understand their impact on performance. **VLM:** Table 5 shows that CLIP encoders with larger parameters perform significantly better, suggesting that they can capture more detailed features, leading to performance improvement. When the LLM is set to GPT-3.5-Turbo, larger parameter CLIP models, specifically ViT-H/14 and G/14, show considerable improvements over the smaller parameter model, B/32, with performance increases of 70.52% and 71.95% respectively. **LLM:** We conduct experiments on CIRCO using various configurations listed in Table 5: no LLM (with fixed templates), an open-source LLM (LLama2-70B), and closed-source LLMs (GPT-3.5-Turbo and GPT-4). The results show that the LLM-based method significantly outperforms the no LLM approach, highlighting the power of dynamic textual reasoning in generating target captions. Furthermore, GPT-based methods perform better than the open-source LLM. Specifically, GPT variants, GPT-3.5-Turbo and GPT-4, achieve 12.12% and 12.75% relative improvements in mAP@5 compared to LLama2-70B, respectively. The plug-and-play design of SEIZE enables the seamless integration of various retrieval and reasoning models, providing flexibility to scale our pipeline as needed and consider the trade-offs between cost and effectiveness, which is crucial when choosing the best LLM and VLM for applications. Thus, we can customize SEIZE to fit any specific retrieval scenario.

## 6 CONCLUSIONS

In this paper, we propose a training-free method called **S**emantic **E**diting **I**ncrement for **ZE**ro-shot composed image retrieval (**SEIZE**), which can utilize off-the-shelf tools to accurately retrieve a specific image based on a reference image and a relative text without training. Our method can consider not only the final similarity between the composed text and retrieved images but also the semantic increment during the compositional editing process. To cover the possible semantics of the composed results, we propose a breadth compositional reasoning to generate diverse edited captions from different semantic perspectives of the reference image. Extensive experiments on three public datasets demonstrate that the proposed SEIZE significantly outperforms the state-of-the-art methods for ZS-CIR. In the future, we will explore semantic editing increment based on LLMs and VLMs in more multi-modal reasoning tasks, such as visual reasoning and conditional generation.

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
