# OpenReview forum: "Semantic Editing Increment Benefits Zero-Shot Composed Image Retrieval"
_acmmm.org/ACMMM/2024/Conference — MM2024 Poster_

### Official Review · Reviewer_NcXU · 2024-05-21

**Rating:** 5
**Confidence:** 3

**Summary:**

In this paper, the authors focused on the zero-shot composed image retrieval. The authors analyzed that the previous studies rely heavily on the robustness of I2T models and text-fusion approaches, which cannot be generalized well in practical applications. The authors proposed a SEIZE method that incorporates a semantic editing search and a breadth compositional reasoning to achieve ZS-CIR. Extensive experiments on three benchmark datasets prove the effectiveness of the proposed method.

**Strengths:**

There are three strong points:
1.	The authors designed a plug-and-play semantic editing search in ZS-CIR, which is novel and effective.
2.	The designed LLM-based breadth compositional reasoning can generate diverse edited captions to improve the model’s robustness capabilities.
3.	Extensive experiments on three benchmark datasets validate the effectiveness of the proposed method.

**Limitations:**

There are also some weak points:
1.	The proposed method involves using MLLM for multiple captions and LLM for reasoning. The introduction of these large models poses challenges for model efficiency. In fact, retrieval tasks demand high responsiveness, so the design of this method may limit its practical application scenarios.
2.	From the experimental results, the performance improvement of the method on the CIRCO and CIRR datasets is quite evident, but the performance improvement on FashionIQ is limited. This may be because the designed method is more inclined towards addressing datasets dominated by modified text or performs poorly on specific domains such as Fashion datasets. The paper lacks further in-depth analysis and explanation of the differences in model performance on different types of datasets.
3.	The authors only conducted ablation study experiments on one backbone and one dataset, lacking persuasive evidence of the effectiveness of different components across various backbones and datasets.

**Suitability:**

3

---

### Official Review · Reviewer_zr9x · 2024-05-24

**Rating:** 4
**Confidence:** 4

**Summary:**

In this paper, the authors studied the zero-shot composed image retrieval task in the field of fashion and in open domain, aiming to explore the semantic edited increment after the composition process based on LLMs, and improve the recall of the task. To address this issue, this paper proposes a semantic editing increment model leveraging multi captions, which first prompts LLMs to generate semantically interactive captions, and then processes semantic incremental correction to change the similarity matrix for retrieval. The experiments show that compared with existing methods, the proposed model can significantly improve the retrieval accuracy. This paper has clear research motivation. However, there are also some problems in writing, which need to be strengthened.

**Strengths:**

a.This paper found that in the process of zero-shot composed image retrieval, the semantic editing increment process has a positive effect on image retrieval, which is the first work to study the zero-shot composed image retrieval task based on utilizing the semantic increment.
b.This paper utilizes LLMs to integrate the multi image captions obtained by pre-trained captioning model and relative captions for composition and employs a measure of semantic increment for improving the retrieval accuracy, which takes advantages of the capability of LLMs and mines the metric relationship in the feature space.
c.Extensive experiments are are carried out on three public data sets, and the experiments showed that the proposed method can improve the performance.

**Limitations:**

a.This paper did not explain the main idea clearly especially for this is a training free process. For example, for the challenge of incorporating the semantic editing increment into the retrieval process, although this paper demonstrated the effectiveness of adding increment before retrieval through certain experiments, it did not clearly explain how the process works from a theoretical perspective, especially when this task did not involve an optimization process to mine such increment.
b.Since this paper proved the effectiveness of each module in CIRCO dataset, but did not provided the ablation experiment details in FashionIQ or CIRR dataset, it is recommended to provide a more detailed ablation experiment in more datasets.
c.There are some colloquial problems in the language of this paper, and some expressions need to be improved. It is suggested to polish the language.

**Suitability:**

3

---

### Official Review · Reviewer_VbGb · 2024-05-24

**Rating:** 3
**Confidence:** 1

**Summary:**

The submission describes an approach to image retrieval heavily depended on large neural networks, that combines text and images in a novel way, so that (less?) training is needed.

**Strengths:**

The paper has a lot of references and tables and the source code of the approach is available. I'm sure another reviewer can make use of it, but I'm too far off this specific field, to judge it.

**Limitations:**

From my point of view - the uninformed reader - the paper is really hard to understand. The interested reader needs to bring along a lot of knowledge in the domain of different ML-based methods to understand the actual point of the paper. Just for the abstract one has to have knowledge about Composed Image Retrieval, Zero-Shot Composed Image Retrieval, image captioning with image2text models, LLMs, and CLIP. Considering this and the list of references, I fear that while the approach is multi-modal, the paper would well fit in ICCV, PAMI, or ICMR rather than ACM MM.

From my perspective I would have a lot more questions, like:
Why would you define similarity? It all sounds so general, but at the end of the day, it's just defined by the test data sets for evaluation, isn't it? Then, the follow up question would be, what is the hypothesis begind the research and the ultimate goal, or how does it translate to applications or further research. I'm missing a graspable und understandable insight.

However, I'm definitely the wrong reviewer for this submission.

**Suitability:**

2

---

### Meta-Review · Area_Chair_EYa8 · 2024-07-03

**Recommendation:** Accept (Poster)
**Confidence:** 5

**Metareview:**

This paper addresses the challenging task of zero-shot composed image retrieval (ZS-CIR) through the SEIZE method, leveraging semantic editing search and breadth compositional reasoning. The proposed approach aims to enhance retrieval accuracy by incorporating semantic increment and diverse caption generation using large language models (LLMs). While the paper demonstrates clear research motivation and achieves non-trivial improvements over existing methods on multiple datasets, there are notable areas requiring improvement: 1) the lack of in-depth theoretical explanation or analysis of the incorporation of semantic editing increment, or any in-depth theoretical analysis of the experimental results (for example, why did it not perform as well on the fashion dataset?), 2) the lack of detailed ablation study, and 3) language and presentation clarity issues.